# COMMUNICATING NATURAL PROGRAMS TO HUMANS AND MACHINES

## ABSTRACT

The Abstraction and Reasoning Corpus (ARC) is a set of procedural tasks that tests an agent's ability to flexibly solve novel problems. While most ARC tasks are easy for humans, they are challenging for state-of-the-art AI. What makes building intelligent systems that can generalize to novel situations such as ARC difficult? We posit that the answer might be found by studying the difference of *language*: While humans readily generate and interpret instructions in a general language, computer systems are shackled to a narrow domain-specific language that they can precisely execute. We present LARC, the *Language-complete ARC*: a collection of natural language descriptions by a group of human participants who instruct each other on how to solve ARC tasks using language alone, which contains successful instructions for 88% of the ARC tasks. We analyze the collected instructions as 'natural programs', finding that while they resemble computer programs, they are distinct in two ways: First, they contain a wide range of primitives; Second, they frequently leverage communicative strategies beyond directly executable codes. We demonstrate that these two distinctions prevent current program synthesis techniques from leveraging LARC to its full potential, and give concrete suggestions on how to build the next-generation program synthesizers.

## 1 INTRODUCTION

Humans solve a range of procedural tasks such as cooking, tying shoes, and programming. Although current AI systems achieve super-human proficiency at certain narrowly specified tasks (Silver et al., 2017; Lerer et al., 2020), their reasoning is domain-specific and fails to generalize to novel situations (Lake et al., 2017). The *Abstraction and Reasoning Corpus* (ARC) introduced by Chollet (2019) presents a set of procedural tasks constructed expressly to benchmark fundamental capacities associated with human general intelligence, including abstraction, generalization, object categories, and the capacity to communicate (Chi et al., 2014; Harlow, 1949; Lake et al., 2017; Lake & Piantadosi, 2020; Bartlett, 1932; Tian et al., 2020; Lombrozo, 2006). Specifically, ARC requires one to infer a procedure consistent with a small number of abstract input-output examples and apply it to a new input to generate an unseen answer, see Figure 1.

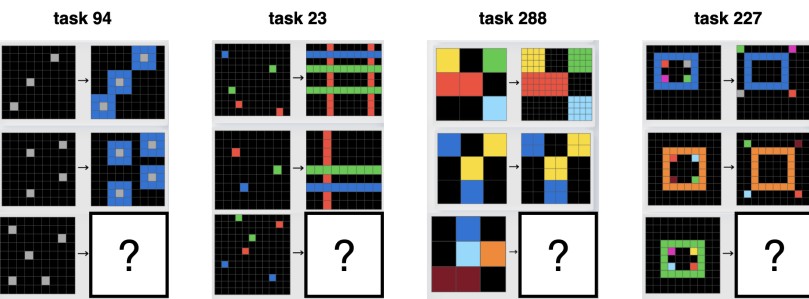

Figure 1: Four ARC tasks, the goal is to correctly infer the unseen output from the given examples.

How do we build systems that are capable of solving general, procedural tasks such as ARC? Traditional approaches of program synthesis (Parisotto et al., 2016; Ellis et al., 2019; Solar-Lezama et al.,

2006; Devlin et al., 2017b) and semantic parsing (Artzi & Zettlemoyer, 2013; Ye et al., 2020; Wang et al., 2015; Marzoev et al., 2020; Guu et al., 2017; Kulal et al., 2019) assume the tasks are **DSL-closed** – aor any task, there exists a program, written in a predefined *Domain Specific Language* (DSL), that solves the task. The ARC benchmark is uniquely *designed* to be **DSL-open** – it does not come with a predefined DSL capable of solving all the tasks. This is both reasonable – most real life tasks, such as cooking and assembling furniture, are DSL-open – and challenging – how can one build an intelligent system that can solve tasks from few examples without a DSL? To illustrate, what might a DSL that would allow one to program all the ARC tasks in Figure 1 look like? This question is difficult to answer; a recent Kaggle competition found that the best AI systems solve at most 20% of the tasks, while Johnson et al. (2021) found that most humans easily solve over 80% [1].

Given that humans greatly outperform the best AI systems in solving ARC tasks, studying the human's cognitive processes (for instance, which set of concepts do human use to represent these tasks?) can shed light on how to build similarly intelligent systems. As these thought processes are not observable directly, we study **natural programs** – instructions that humans give to each other, as a window into these latent cognitive processes. Like computer programs, these instructions can be reliably interpreted (by another human) to produce the intended output. Unlike computer programs, which must be stated in a specific style, natural programs can be stated in any form – such as verbal instructions or input-output examples – as long as another human can execute them. In this work, we study a particular form of natural programs, that of *natural language instructions*. We show that analyzing these natural programs – with explicit comparisons to computer programs – can both shed light on how humans communicate and interpret procedures (Spelke et al., 1992; McCarthy et al., 2021; Clark et al., 1983; Clark & Wilkes-Gibbs, 1986) and inform how one may build AI systems for challenging, DSL-open domains such as ARC.

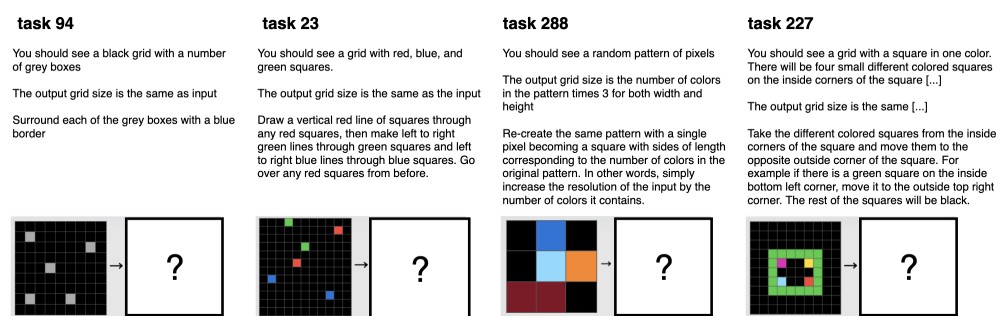

Figure 2: Four LARC tasks, corresponding to those of Figure 1. The goal is to produce the correct output given *only* the language instructions. 88% of the ARC tasks can be communicated this way.

We present the **Language-complete Abstraction and Reasoning Corpus** (LARC) of natural language instructions elicited from a two-player communication game, where 88% of the ARC tasks can be successfully communicated. LARC tasks are **language-complete**: The successful instructions contain all the relevant information, even in absence of the original input-output examples (see Figure 2). This is important in several ways: First, one can use LARC to study how humans use language to communicate abstract procedures, as humans clearly have the capacity to both *generate* and *execute* these natural programs; Second, one can directly see what concepts an intelligent system must be aware of (such as colors and numbers); Third, as people readily generate natural programs, studying them will provide insights on building interactive systems.

We perform **linguistic analysis** on LARC, finding that humans readily leverage algorithmic concepts without being explicitly instructed to do so. These concepts range from domain general ones, such as loops, to domain-specific concepts such as flood-fill. However, natural programs in LARC are distinct from typical computer programs in two ways: (1) They contain a wide range of concepts, in contrast to the few primitives in a typical DSL; (2) The majority of effort is spent on providing clarifications and validations, as opposed to stating the procedure verbatim. We apply standard **program synthesis** algorithms on LARC, finding that while existing approaches can benefit from the additional language annotations, the two aforementioned distinctions pose significant challenges

---

[1]Humans were evaluated on a subset of the training tasks; the Kaggle competition used a private test set.

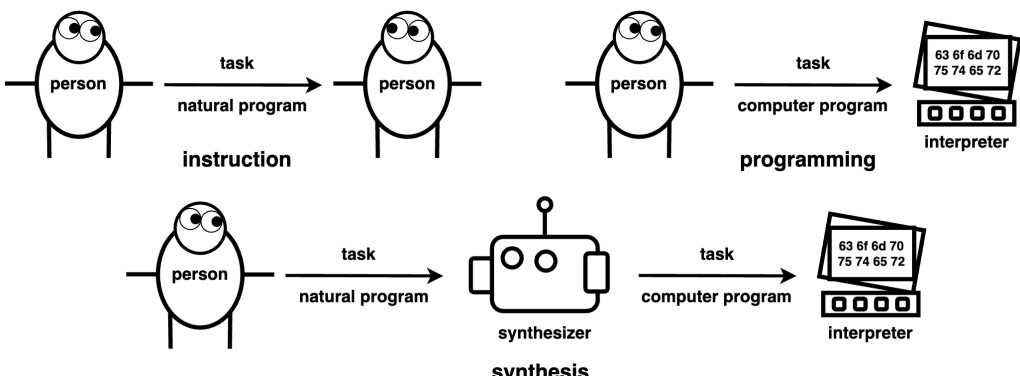

Figure 3: Three ways humans and machines communicate with each other: instruction (top-left), programming (top-right), synthesis (bot).

## 2 COMMUNICATING AND INTERPRETING PROGRAMS

In communicating procedures, a *programmer* constructs a *program* in a suitable language, which is then executed on an *interpreter*, producing a desirable behaviour. For instance, a person can *instruct* another person to carry out a certain task (Fig. 3 top-left), or directly *program* a machine to solve tasks using code (Fig. 3 top-right). Program synthesis takes in an instruction, and reformulates it as executable code, insulating the person from the programming process (Fig. 3 bot).

How do we build machines that are capable of solving challenging tasks given in a natural way? Typically, one follows a "DSL-first" approach. In the DSL-first approach, one first defines a programming language and builds a corresponding interpreter, so that a skilled programmer may express the tasks as programs. Then, one naturalizes the initial DSL with a synthesizer, allowing end-users to describe tasks using language (Artzi & Zettlemoyer, 2013; Artzi et al., 2014; Ye et al., 2020; Wang et al., 2015; 2016; Marzoev et al., 2020), or by giving examples (Ellis et al., 2019; Solar-Lezama et al., 2006; Pu et al., 2020). While this DSL-first workflow has yielded impressive results, the DSL itself is also a single point of failure. It is difficult to design DSL with the right *scope*, so that it is easy to write programs for the domain, without bloating the language with redundant concepts (Gulwani et al., 2015; Bruce et al., 2020; Soto-Valero et al., 2021). For synthesis systems, one must ensure that the DSL *aligns* reasonably to human instructions (Liang, 2016; Shin et al., 2021), while simultaneously being *efficient* when used by the synthesizer (Ellis et al., 2019; 2020). These challenges may explain why ARC, and other DSL-open domains, pose significant challenges in building intelligent systems using the DSL-first approach.

In this work, we suggest that *starting* with human instructions (Fig 3 top-left) can yield insights for building systems on challenging, DSL-open domains and highlight deficiencies in DSL-first approaches. We define a **natural program** as instructions constructed by a person that can be interpreted by another person to produce a specific output. This program is *natural*–it can be understood by speakers of the language[2] without a prior consensus–but behaves as a *program*, in that it produces a definitive output, which can be unambiguously checked for correctness. For instance, the original ARC (Chollet, 2019) tasks are natural programs: Given a program consisting of input-output examples, a fellow human can readily interpret this program to produce an output on a new input, which can be checked for correctness. Contrary the DSL-first approach, by starting with (linguistic) natural programs, one can directly observe the set of concepts and strategies necessary to master a domain (such as ARC), without having to first build an interpreter.

---

[2]language here is to be understood loosely as any medium of communication between people

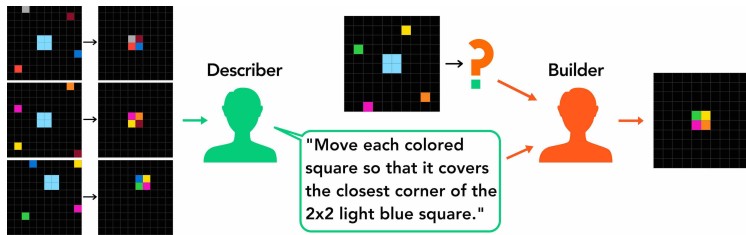

Figure 4: a **describer** instructs a **builder** how to solve an ARC task using a natural program

# 3   LARC: LANGUAGE-COMPLETE ABSTRACTION AND REASONING CORPUS

We present a dataset that augments the original ARC tasks from Chollet (2019) with *language-complete* instructions: they can be demonstrably interpreted by other human users to correctly produce the intended outputs without any additional contexts (including the original input-output examples). Thus, LARC tasks (Fig 2), like their counterparts in ARC, meet the definition of natural program while containing only natural language descriptions. To collect this dataset, we introduce a *communication game* were human describers produce linguistic instructions for unseen downstream human builders asked to solve the same tasks; and deploy this experiment using a novel bandit algorithm to efficiently collect verifiable natural programs. The final dataset augments 88% of the original ARC tasks (354/400) with at least one verifiable *natural program* description that could be successfully interpreted by another human participant to solve the task. Fig. 5(C-D) shows the distribution of success rates for participants acting as describers and builders over time.

## 3.1   HUMAN ANNOTATION DETAILS

We recruited 373 subjects via Amazon Mechanical Turk who were paid for 45 minutes of work. Fifty individuals were excluded for failing to complete the task, so the final analysis included 323 subjects. The study was approved by our institution's Institutional Review Board, did not collect personally identifiable information, and did not pose risks to participants. Subjects were paid $6.00 and a $0.25 bonus for every successful communication. Subjects averaged 5.5 communications, bringing their expected hourly wage to $9.83. For interface and consent form see Appendix A.2.

## 3.2   TWO-PLAYER COMMUNICATION GAME

For each task, a participant may be assigned one of two roles: a **describer** or a **builder**. The describer plays the role of a *human synthesizer*, who reformulates input-output examples (of ARC) to natural language descriptions. The builder plays a role of a *human interpreter*, who must construct the correct output on a new input without access to the original examples (Fig 4). The description is structured into three sections to incentivize consistency: (1) what the builder should expect to see in the input, (2) the output grid size, and (3) what the builder should do to create the output (Fig 2). After the description was submitted, we verify the describer's own understanding by asking them to build it, and discarding the submission if the describer fails. The describer was shown all previous verified descriptions for a task, alleviating challenge of solving the task from scratch. Participants construct the output using actions defined in ARC, such as `paint(color,x,y)`, `copy/paste`, and `floodfill`. All action sequences are recorded.

## 3.3   THE BANDIT ALGORITHM FOR DATA COLLECTION

Collecting valid linguistic natural programs requires significant human efforts: For each task (of varying difficulties), natural programs must first be *proposed* by a number of describers, and then *validated* by a number of builders, where both can make mistakes. Thus, A naive data-collection process that simply collects a fixed number of descriptions and builds per task will be expensive. To address this challenge, we formulate the following bandit problem: *multi-bandit* – each of the 400 ARC tasks is a different bandit; *infinite-arm* – given a task, each natural language description (there are infinitely many) is a different arm; *best-arm identification* – once a natural program is proposed, we must validate it. We develop a novel bandit algorithm (Appendix B) to solve this problem, as to

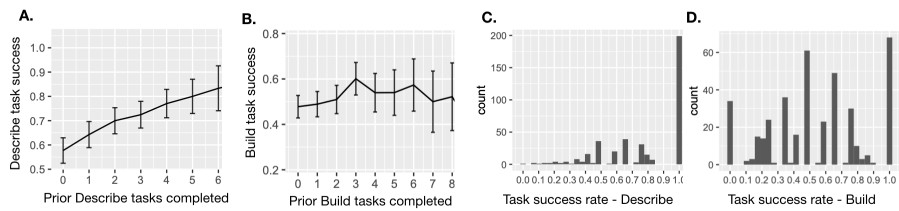

Figure 5: *A.* Describer improves at verifying their own descriptions as a they describe more tasks. *B.* Builders do not improve at constructing the correct outputs as they build more tasks (likely due to having no control over the qualities of their given descriptions). *C.* Rate of describers verifying their own descriptions (avg 75%). *D.* The rate of builders constructing the correct output, (avg 50%).

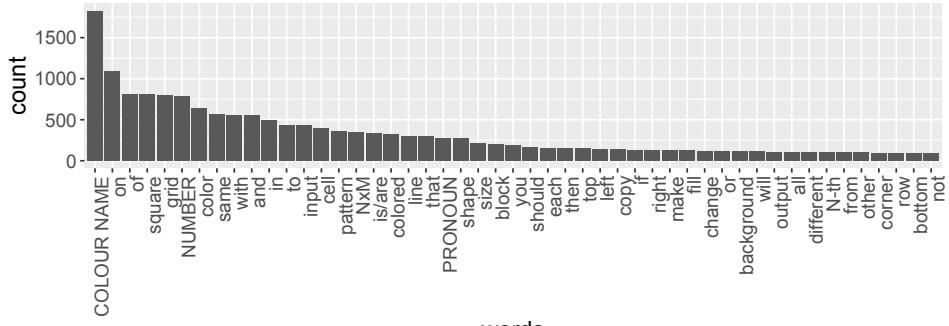

Figure 6: Words used in successfully built descriptions, sorted by their frequency in the corpus (total 642 unique words). The words were singularized. Colors names, numbers, and pronouns were grouped together.

our best knowledge, no known algorithm can be directly applied. For each MTurk participant, our bandit algorithm dynamically allocates a set of describing and building efforts for their session. As a result, the LARC dataset was annotated for $3667, whereas a naively collecting 20 annotations per task would cost at least $10,800.

## 4 LINGUISTIC ANALYSIS OF NATURAL PROGRAM PHRASES

How do people actually use language in LARC to produce robustly interpretable descriptions for each task? In this section, we describe results from a manual *linguistic analysis* of verified natural program phrases – phrases contained in descriptions that resulted in a successful build. We analyse the verified phrases under the lens of *computer programs*, highlighting certain aspects that are shared between computer and natural programs, and other aspects that are distinct. Our analysis codes individual phrases with manual *tags* corresponding to general concepts from computer programs and concepts from *core knowledge* (Spelke & Kinzler, 2007). In total, we manually label 532 randomly sampled phrases (22% of the phrase corpus) using 17 conceptual tags (in which multiple tags can be applied to each phrase); Figure 7A. shows a frequency of these tags. For details see Appendix A.3.

### 4.1 SIMILARITIES OF COMPUTER AND NATURAL PROGRAMS

**General Algorithmic Concepts**   We find LARC contains algorithmic concepts similar to those that can be found in a typical programming language (i.e. python, javascript). For instance, **tag_logic** details a specific condition (i.e. "the box is blue"), **tag_array** references a group of similar objects (i.e. "you should see four red shapes"), and **tag_loop** is similar to while loops ("keep going until ..."). Note that humans readily generate and execute these concepts without being directly instructed to do so, suggesting that humans reason about ARC tasks algorithmically.

**Domain Specific Concepts**   Like computer programs written in a specific library, we find that LARC contains concepts that distinguish it from other domains. We focus on concepts based on **core knowledge** (Spelke & Kinzler, 2007), specifically, the object system defined by the principles

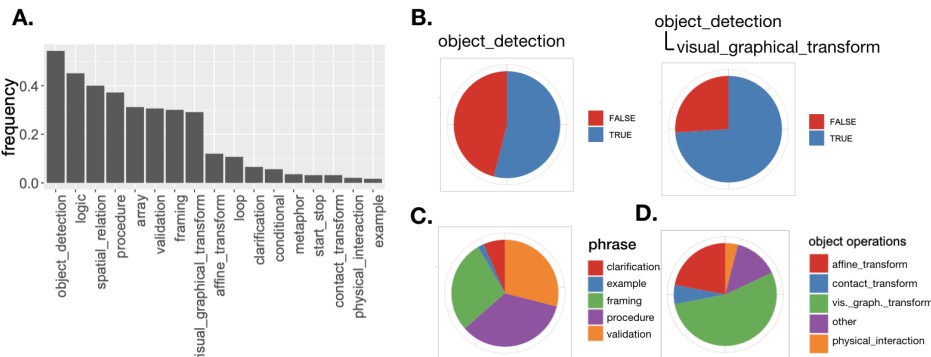

Figure 7: **A.** The frequencies of all tags occurring in human phrases. Each phrase can have multiple tags. **B.** More than half of the phrases described objects, of which, 75% described spatial relations. **C.** Relative frequencies of code (procedures) against non-code (example, framing, clarification, validation). **D.** Relative frequencies of core knowledge topics in phrases that referenced objects.

of *cohesion, persistence*, and *influence via contact*, which the ARC corpus was designed to leverage – " The ARC priors are designed to be as close as possible to Core Knowledge priors, ... a fair ground for comparing human intelligence and artificial intelligence (47).". We found that about a half of the phrases referenced **objects**, and three quarters of them described spatial relations of one object to another (Fig.7B). Fig.7D shows the relative frequencies of operations performed on objects. While objects are prevalent in LARC, most operations on objects were of the category **visual_graphical_transform** – such as recolouring and creating new objects. In contrast, only 5% of phrases could also be read as influence via **physical_interaction**. This suggests that objects behaviours in ARC are rooted more in abstract graphical operations, possibly because it is difficult to represent realistic physical interactions in the input-output format.

## 4.2 Differences of Computer and Natural Programs

While natural programs in LARC are algorithmic, certain aspects remain markedly different from computer programs. We outline two (related) ways natural programs differ from computer programs. First, instead of using a narrow DSL with few primitives, natural programs use a large, diverse set of primitive functions. Second, instead of stating a precise procedure verbatim, natural programs rely on a range of additional strategies to ensure that they can be interpreted precisely.

**Natural Programs Invoke a Large Number of Concepts** Since LARC is language-complete, analyzing the words used in LARC serves as a good proxy for the underlying concepts present in the ARC domain. Similar to the finding of Johnson et al. (2021), we find that humans use a wide range of concepts (Fig 6). This is a testament of the general capabilities of the human interpreter: the builders readily execute these concepts, and the describers confidently invoke these concepts to a stranger (the builder). LARC also suggests that, to engage with an interpreter capable of executing a wide range of concepts, the task of recalling the right concept at the right time will be nontrivial. While it is sufficient to recall a concept by a unique name such as 'block' for a narrow DSL, it is evident from LARC that even the same concept will have a range of different names such as 'square' or 'box' or 'pixel', all of which can be correctly interpreted under the right context.

**Natural Programs Communicate Information Beyond Procedures** We study the relative frequencies of directly executable commands – **tag_procedure**, in contrast to "meta information", which can be analogized roughly as: **tag_framing** – comments and setting up which library to use, **tag_validation** – checks and assertions, to ensure correct execution, and **tag_clarifications** – restating the same procedure in different words. Notably, only about a third of the LARC phrases are procedures, while the framings and validations occur at roughly the same frequency (see Fig.7 C). In contrast, 86% of the executables in computer programs are not commented (Huang et al., 2020).

The high frequency of framing tags in LARC suggests that describers carefully establish context to resolve uncertainty over which programmatic concepts may be relevant, anticipating the large number of concepts that the builder may have in mind. We also find that the describer often assumes

the directly executable portion of a natural program (i.e. tag_procedure) as inherently *ambiguous*, as suggested by frequent use of tag_validations and tag_clarifications following these procedures. Specifically, validation is meant to actively help the interpreter in choosing a correct interpretation among multiple plausible interpretations. Clarification amends the initial ambiguous explanation, with another, also ambiguous explanation, resulting in narrowing the distribution of possible interpretations. We posit that a better analogy to natural programs are not computer programs, but program synthesis/inference, where an intelligent agent actively searches for the right interpretation among many alternatives given the natural program (of any form) provided by the describer.

## 5 EXECUTING NATURAL PROGRAMS

We evaluate whether DSL-first program synthesis methods (Fig 3, bot) can execute natural programs as well as humans do. We consider three kinds of natural programs: (1) Input-output examples from the original ARC corpus (IO); (2) IO in conjunction with successful language instructions in LARC (IO+NL); And (3) language alone (NL-only) – same as the MTurk builder task.

### 5.1 PROGRAM SYNTHESIS

In (symbolic) program synthesis (Solar Lezama, 2008; Devlin et al., 2017b; Guu et al., 2017), the synthesizer takes in a natural program, and reformulates it as code over a DSL, which can be directly executed on an interpreter. We have manually crafted a DSL based loosely on the concepts present in the LARC corpus and built its corresponding interpreter (see Appendix A.4).

**Generate and Check Using IO**  If the given natural program contains IO examples, the standard symbolic program synthesis approach (Solar-Lezama et al., 2006; Devlin et al., 2017b) follows the *generate and check* strategy. Let $natprog$ be a natural program, the synthesizer returns programs $prog$ from a DSL from the following distribution:

$$P_{synth}(prog|natprog) = P_{gen}(prog|natprog)\mathbb{1}[prog \vdash IO]$$

$P_{gen}$ is the generative distribution: given a natural program, it proposes program $prog$ from the DSL. $\mathbb{1}[prog \vdash IO]$ is the checker: it validates $prog$ by executing it on the interpreter, ensuring that $prog(x) = y$ for all input-output pairs $(x, y) \in IO$. The key strength of this approach lies in its generalizability: If a proposed program can be checked against all IO examples, it is very likely to generalize to an new instance of the same task due to the inductive bias of the DSL.

**Generation Models**  Our $P_{gen}(prog|natprog)$ generates programs in two parts: a neural model outputs a tree bigram over the grammar of the DSL (Lari & Young, 1990), then a dedicated Ocaml enumerator deterministically enumerates programs from a PCFG fitted to this bigram distribution in decreasing probability (Ellis et al., 2020). For simplicity, we report results of unconditioned generators $P_{gen}(prog)$ (i.e. a fitted prior) when language is absent, and language-conditioned models $P_{gen}(prog|NL)$ when language is present. This way, we can use the same $P_{gen}(prog|NL)$ model for both IO+NL and NL-only tasks in the test set, as it does not depend on IO. Similar to (Dechter et al., 2013; Catherine et al., 2021), we first bootstrap our generative models with 10 "seed" programs, discovered by enumerating an uniform unigram model. We find more complex neural models using a CNN encoder do not bring significant benefits, and a neural sequence decoder (as in Guu et al. (2017)) performs poorly (for details see Appendix A.5).

**Leveraging Language**  We use a pre-trained model (T5, Raffel et al. (2019)) to represent language by taking an average of its encoded tokens. To encourage the learning of compositional relationships between language and program, we use **pseudo-annotation**, similar to recent methods that have leveraged synchronous grammars (Marzoev et al., 2020; Jia & Liang, 2016; Shin et al., 2021; Catherine et al., 2021). First, we provide linguistic comments for each primitive function in the program DSL (e.g. `flood_fill(color)` with *fill with the color*). Then, during training, we obtain additional paired language and program examples by substituting primitives of artificial programs with their corresponding comments [3]. For more examples of pseudo-annotation see Appendix A.4.

---

[3]for instance, (`lambda (to_original_grid_overlay (remove_color(grid_to_block x) yellow) false)`)  becomes *place block on input grid remove color from block yellow*

| training tasks discovered | | |
|---|---|---|
| | no-pseudo | pseudo |
| IO | 15 / 200 | - |
| IO + NL | 13 / 200 | **21 / 200** |

| testing tasks solved | | |
|---|---|---|
| | no-pseudo | pseudo |
| NL-only | 1 / 183 | 0 / 183 |
| IO | 18 / 183 | - |
| IO + NL | 16 / 183 | **22 / 183** |

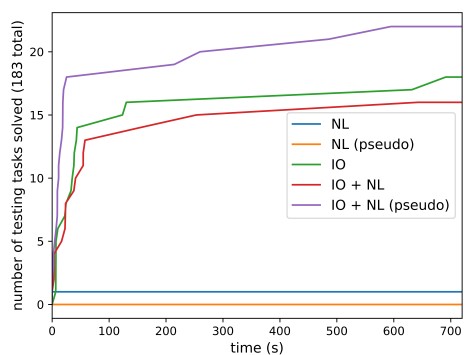

Table 1: Executing different kinds of natural programs (IO – Input-output examples from the original ARC corpus, IO+NL – IO in conjunction with successful language instructions in LARC, NL-only – same as the MTurk builder task) using program synthesis. Training tasks discovered under distant supervision (top). Testing tasks solved (bottom). Models trained with NL can additionally augment their training with pseudo-annotations consisting of artificial language-program pairs.

Figure 8: Number of test tasks solved for the three kinds of natural programs, IO, NL, IO+NL, with and without pseudo annotations, as a function of enumeration time. Of all models we consider, IO+NL+pseudo achieves the best (22/183) outcome. There are error bars as the bigram enumerator is deterministic. It is possible (but not likely) that re-training these models will have an effect due to the randomness of sampling pseudo-annotated programs.

**Distant Supervision** LARC, similar to SCONE (Long et al., 2016), falls under the challenge of distant supervision: each training task only contains the correct output, but not the ground-truth program responsible for generating it. We adopt the iterative approach used in (Dechter et al., 2013; Ellis et al., 2020; Catherine et al., 2021; Guu et al., 2017) to *discover* suitable programs during the training phase. Specifically, we alternate between (1) searching for solutions by enumerating from bigrams proposed by our current neural model per task, (2) using the discovered programs to fit better bigram distributions that maximizes the marginal likelihood of the discovered programs per task, and (3) using these additional bigrams as targets to train our neural model.

## 5.2 RESULTS

We split the ARC tasks into 200 training tasks and 200 testing tasks, and additionally filter the testing tasks down to 183 tasks with successful NL communications. We then train the models for a wall clock time of 10 hours each using iterative learning. We test on the 183 test tasks by first using the neural model to propose a bigram per task, then enumerating the bigram for 720 seconds. We keep the top-3 most likely programs that also satisfy the IO examples if the natural program contains IO. We then check if any of the top 3 programs satisfies test input-output. See Table 1 and Figure 8.

**Quantitative Findings** Our best model, IO+NL+psuedo, solves only 22/183 of the training tasks. We believe this model obtains better results than IO+NL (22 vs 16) likely due to the lack of training data in the LARC corpus (only hundreds of NL annotations) whereas with psuedo-annotation one can generate an infinite number of artificial NL-prog pairs, albeit limited to how well each primitive's comment resembles that of the language descriptions in LARC. We note that having the ability to *check* if a proposed program is correct under IO is *crucial* for the success of current program synthesizers, with no more than 1 task solved with NL-only – Like the validation phrases in LARC, the input-output examples in IO serve as a form of *validation* for the enumerative synthesizer. This corroborate with similar findings in Austin et al. (2021).

**Qualitative Findings**   We investigate in what way does language affect synthesis. We compare the bigram model conditioned on language (IO+NL+psuedo) to the unconditional bigram model (IO). Specifically, for each primitive in our DSL, we ask how many times more likely is it going to appear in correct programs generated with the language-conditioned bigram vs the unconditioned one? We plot this ratio on a log scale for all primitives that were used in ground-truth programs, see Figure 9. We note that for most primitives that are frequently used, the language-conditioned generator is more likely to generate the correct primitives than the unconditioned generator. We conclude that while language definitely helps current approaches, the overall results are still poor.

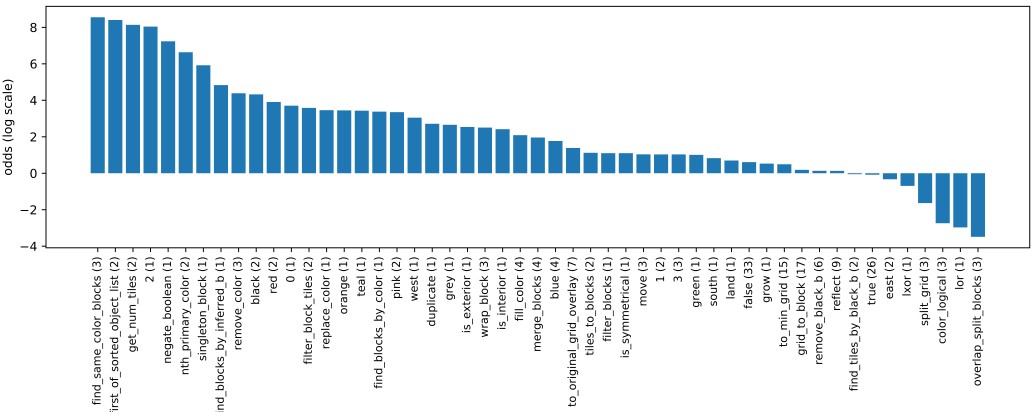

Figure 9: Relative odds of using the correct primitive for a task, language-conditioned vs unconditioned generation. Number in parenthesis denotes the total number of times a primitive is used.

**Challenges**   We highlight key challenges from our experience in executing natural programs. *Scoping*: We are never sure if our current DSL can explain all the tasks in a DSL-open domain, leading to a cycle of continuously adding more primitives and refactoring. *Referencing*: How might a synthesizer reliably attend to the relevant concepts from a large number of irrelevant concepts (that are required to operationalize rest of ARC)? *Not Paraphrasing*: When natural programs are NL-only, our best model solves only 1 / 183 tasks because it, like other NL-to-code approaches, assumes a close, 1-to-1 paraphrasing relationship between language and code, leading it to miss crucial *framing* and *validation* statements, which occurs in abundance in natural programs.

**Suggestions**   It is likely that any intelligent system in a general domain will have to operate with a large number of concepts. Thus, if we choose DSL as a representation of concepts, we need to grapple with the challenge of extending and synthesizing a large DSLs. Alternatively, program *induction* (Gaunt et al., 2016; Devlin et al., 2017a) might allow an agent to represent concepts without an explicit DSL given the right data. Recent advances in large language models, such as codex, (Chen et al., 2021) have demonstrated conventional understandings of language and programs. We expect that specific domains, such as LARC, may benefit from adapting conventional understandings. Finally, approaches that leverage different communicative strategies in linguistic instructions (Sumers et al., 2020) will be crucial in handling NL-only instances.

## 6   CONCLUSION

We propose **natural programs** – instructions that can be interpreted by humans to produce a verifiable output – as a pathway to building intelligent systems that can solve general tasks given in a natural way. To this end, we present the *Language-complete Abstraction and Reasoning Corpus* (LARC). By being simultaneously *DSL-open* and *language-complete*, LARC highlights the distinctive gap between communication to humans and machines: Where programmers struggle to build a program synthesizer, end-users readily communicate the same tasks using words alone. We hope that by studying and curating the LARC dataset multiple communities – such as psychology, cognitive science, and machine learning – can work together to close this gap.

POTENTIAL NEGATIVE IMPACT

The long-term goal of this work is to 'reverse-engineer' how humans think and communicate, in order to jointly inform cognitive research – how do people solve and learn from structured tasks – and computational systems, that can learn and collaborate with humans. While this work takes constrained steps in these directions, it assumes both of these aims as important end goals. Any system that seeks to accurately use and interpret human language raises concerns regarding safe deployment for downstream applications, for instance, non-experts operating safety-critical equipment using natural language. Further, AI could exploit ambiguity in human language to write legally-binding agreements that can be read in different ways.

REPRODUCIBILITY STATEMENT

The source code for the models and the datasets are available at the following github repository, along with instructions on how to use them: `https://anonymous.4open.science/r/ec-A3EE`

The bandit web code that collects human responses, and the LARC dataset itself, is given in the suppliment ZIP file.

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

## A    APPENDIX

The appendix serves as a complimentary document to the paper detailing the data collection process, analysis, and program synthesis. It should be used in conjunction with the following:

1. the LARC dataset and its annotation workflow, and bandit algorithm can be found in the supplementary zip file `LARC`.

2. program synthesis using language codes is at this URL :
   `https://anonymous.4open.science/r/ec-A3EE`

### A.1    THE LARC EXPLORER GUI

We encourage the reader to explore the dataset first-hand using the explorer GUI (Figure 10):

1. point to the `LARC` root directory

2. run `python3 -m http.server`

3. open `localhost:8000/explore/` in a chrome browser

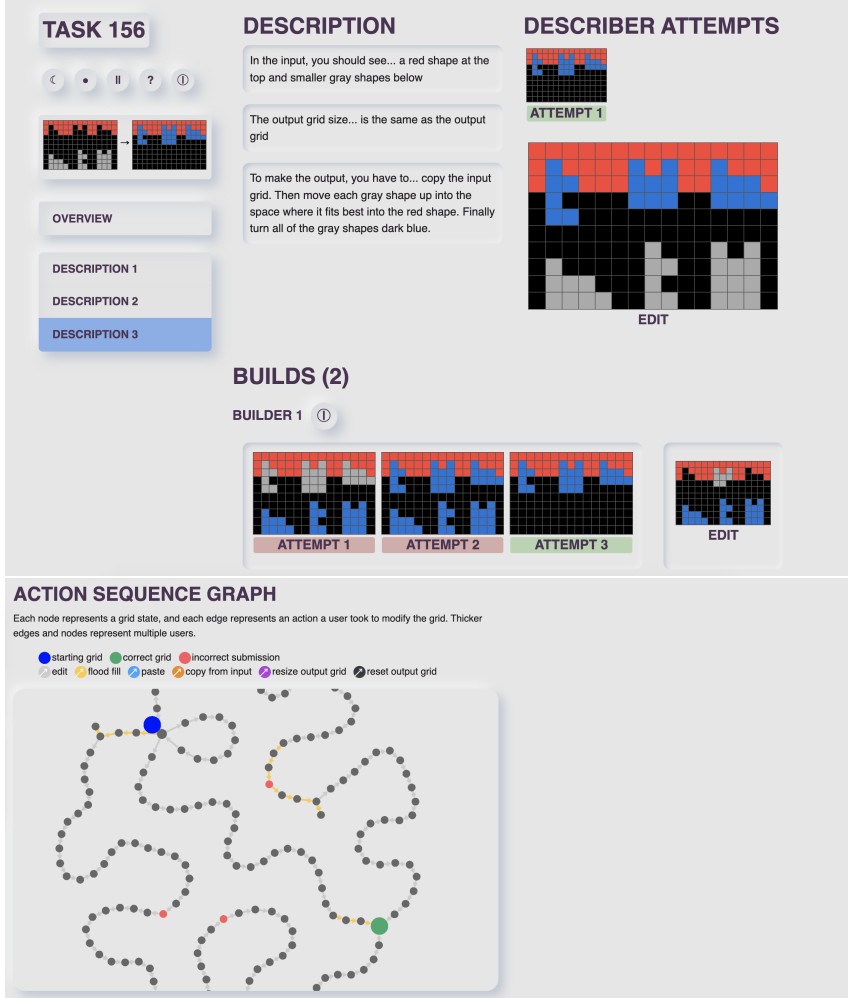

Figure 10: The explore interface for task 156 (top). action sequence graph of builder 1 (bot)

## A.2 CONSENT FORM AND ANNOTATION WORKFLOW

**Consent Form** In this study, you will interpret descriptions of an abstract pattern that you observe in grids. By answering the following questions, you are participating in a study performed by cognitive scientists in [*author institution*]. If you have questions about this research, please contact [*author*] at [*author email*]. Your participation in this research is voluntary. You may decline to answer any or all of the following questions. You may decline further participation, at any time, without adverse consequences. Your anonymity is assured; the researchers who have requested your participation will not receive any personal identifying information about you. By clicking 'I AGREE' you indicate your consent to participate in this study.

**Annotation Workflow** Then, the user is given tutorials about communicating ARC tasks, and dynamically assigned a sequence of describe and/or build tasks until they have completed 45 minutes of work. Figure 11 shows the build and describe interface. For full workflow see `LARC/collection`.

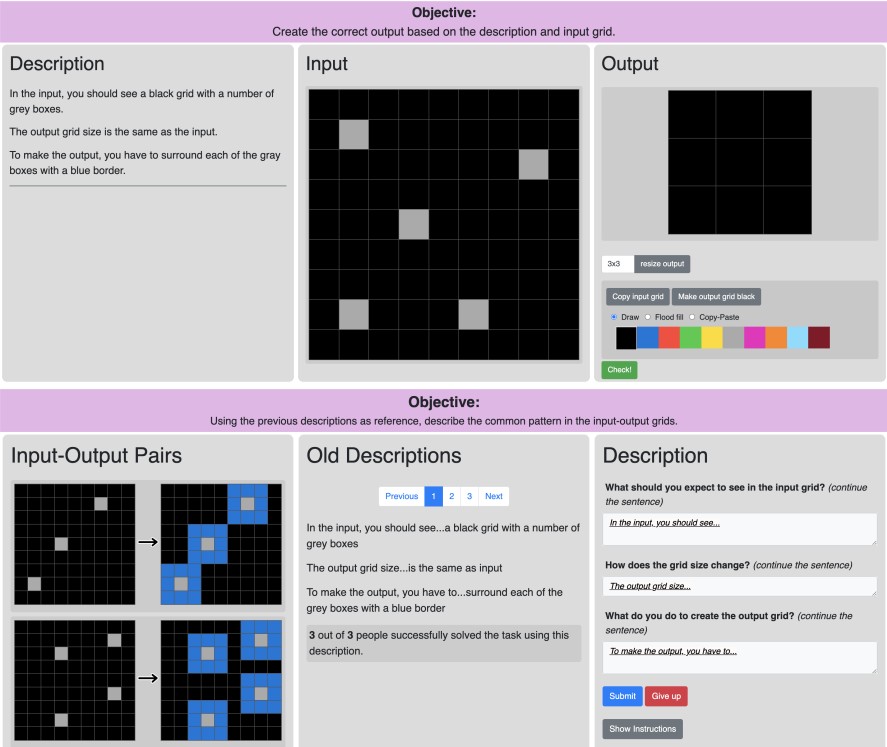

Figure 11: *A*. The builder interface. *B*. The describer interface.

## A.3 LARC Linguistic Analysis Tagging Scheme

The tagged phrases can be found at `LARC/dataset/annotated_phrases.csv`

The phrases were codified by expert coders using a set of 17 binary tags. For each tag, a phrase can either be a positive instance (+) or a negative instance (-) [4]. The following table details the tags and coding scheme used:

| Tag | Description | Examples |
|---|---|---|
| **Procedure** | Directly commands the builder to do something; If you were to delete it, the program will fail to execute. | (+) Fill each enclosed hole with yellow
(-) look at the color that form the design in the input. |
| **Metaphor** | A metaphor can be an analogy or reference to human common sense knowledge – e.g. spiral. | (+) A random green pattern
(+) A pattern like a long A |
| **Clarification** | A phrase made following a previous statement that attempts to clarify misinterpretations. | (+) Then, copy and paste each colored square in the input grid 4 times – once in each "quadrant"
(+) (or 5 rows or whatever the number of rows is before it repeats).
(+) Where there's a dark blue square, put orange squares directly above and below it (4 total). |
| **Example** | Gives a concrete instance. | (+) The opposite is also true (for example if it is light blue, change to dark red). |
| **Array** | Makes a comment about a collection of objects sharing some common property. | (+) Where there's a dark blue square, put orange squares directly above and below it (4 total).
(+) Leave the magenta and light blue squares as they are; do not add anything to them if they are present. |
| **Validation** | After the builder executes a procedure, check if they got the right answer (i.e. asserts, test-cases, verification, or error handling). | (+) You should end up with all blue boxes touching each other
(+) Fill in all of the black boxes to complete the pattern until there are no more black boxes. |
| **Loop** | Includes a looping procedure, such as the use of *while*, *for*, *until*, *for each*, or *repeat*. | (+) Continue coloring green until you reach the center of the grid.
(+) Reduce the grid size so that one square is available for each group. |
| **Start_Stop** | Talks about the process or duration of some operations. | (+) start at the upper right corner
(+) the red shape needs to move until it is touching the blue cube |
| **Conditional** | Of the form *if X then Y*. | (+) If they do not match, make the output square green. |
| **Logic** | Includes first-order logic, such as *same*, *and*, *or*, or *not*. | (+) The same size as the input (+) You will not use dark blue squares at all (-) A 4x4 pattern |
| **Framing** | Sets up the problem by offering a particular point of view, defining some objects to be referred to later. | (+) four colored area.
(+) 1 or 2 squares filled in with the same color on a black background. |

---

[4]marked by 1 and 0 respectively in the csv

| Tag | Description | Examples |
|---|---|---|
| **Spacial Relation** | Any reference to a relative position in space to some other component. Positive examples include: under, reaches, touches, angle, outer, downward, parallel, near, after, in between, central, etc. | (+) The red shape next to the blue shape
(+) Put yellow inside the green |
| **Physical Interaction** | Any reference to an imaginary force. | (+) The red object falls
(+) Blue slides to the left towards red |
| **Contact Transform** | Influence via contact, i.e. any specialized version of physical interaction that involves *at least two objects* and *some type of contact causality*. | (+) Move X until contact with Y
(+) Set X touching Y and turn it the color of Y
(-) Red moves left one square |
| **Affine Transform** | Any reference to a affine transformation over an object, such as rotation, translation, etc. | (+) Rotate 90 degrees
(+) Extend the square into a line |
| **Visual-Graphical Transform** | Any other visual or graphical modification other than a geometric one, such as coloring, flood-fill, or drawing a new shape. | (+) Make it gray
(+) Draw a line |
| **Object Detection** | The localization of a cohesive, bounded object. | (+) The red shape
(+) Move **it** to the left
(+) The pattern |

These tags can also be grouped hierarchically into the following categories:

**Programmatic:** procedure, array, validation, loop, start_stop, conditional, logic

**Human/Mechanisms for Domain General Communication:** metaphor, clarification, example, framing

**Objects and Object Manipulation:** spacial_relation, physical_interaction, contact_transform, geometric_transform, visual_graphical_transform, object_detection

## A.4    THE ATTEMPTED LARC DSL

As LARC is DSL-open, we must first *construct* a suitable DSL before applying (symbolic) program synthesis approaches. Here is our attempt at constructing such a DSL. For each DSL primitives, we also list its corresponding pseudo-annotation comments. We hand-designed DSL a for the LARC domain consisting of 103 primitives (implemented as a set of polymorphically typed $\lambda$-calculus expressions) intended to be broadly and basically applicable to all tasks on the domain – the DSL operates over grids of pixels, and contains simple functions designed to repeatedly perform image transformations over pixel grids to produce an output grid. The complete DSL is available at the released code repository; below we provide representative example functions and the accompanying natural language glosses of their behavior used in the *pseudoannotations* generative procedure; as well as sampled program expressions and their generated pseudoannotations.

| **Example DSL Functions and Natural Language Gloss Function Annotations** | |
|---|---|
| *DSL Function* | *Natural Language Gloss* |
| blocks_to_original_grid | 'place blocks onto input grid' |
| blocks_to_min_grid | 'get the smallest grid containing the blocks' |
| first_of_sorted_object_list | 'get the block with the smallest or greatest value of' |
| singleton_block | '' |
| merge_blocks | '' |
| filter_blocks | 'remove the blocks that have' |
| map_blocks | 'for every block' |
| filter_template_block | 'find the main block' |
| reflect | 'reflect' |
| move | 'move' |
| center_block_on_tile | 'move block to tile' |
| duplicate | 'duplicate' |
| grow | 'enlarge' |
| fill_color | 'color the block' |
| fill_snakewise | 'color the block in a snake pattern with' |
| replace_color | 'replace colors' |
| remove_black_b | 'remove the black background' |
| remove_color | 'remove color from block' |
| box_block | 'get smallest rectangle containing block' |
| wrap_block | 'surround block with' |
| filter_block_tiles | 'only keep tiles that' |
| map_block_tiles | 'for each tile of block' |
| to_min_grid | '' |
| to_original_grid_overlay | 'place block on input grid' |
| get_height | 'get height of block' |
| get_width | 'get width of block' |
| get_original_grid_height | 'get the height of the input grid' |
| get_original_grid_width | 'get the width of the input grid' |
| get_num_tiles | 'count the number of tiles of the block' |
| nth_primary_color | 'find the nth most common color' |
| is_symmetrical | 'is the block symmetrical' |
| is_rectangle | 'is the block a rectangle' |
| has_min_tiles | 'does the block have at least n tiles' |
| touches_any_boundary | 'does the block touch any edge of the grid' |
| touches_boundary | 'does the block touch the edge' |
| has_color | 'does the block have color' |
| is_tile | 'is the block a tile' |
| block_to_tile | '' |
| get_block_center | 'get the central tile of the block' |
| map_for_directions | 'in every direction' |
| find_same_color_blocks | 'find blocks based on shared color' |

| | |
|---|---|
| find_blocks_by_black_b | 'find blocks based on if they are separated by the black background' |
| find_blocks_by_color | 'find blocks based on if they are separated by the given color background' |
| find_blocks_by_inferred_b | 'find blocks based on if they are separated by the background' |
| grid_to_block | '' |
| split_grid | 'split the grid in half' |
| find_tiles_by_black_b | 'find the tiles based on if they are separated by the black background' |
| is_interior | 'is the tile in the interior of a block' |
| is_exterior | 'is the tile in the exterior of a block' |
| tile_touches_block | 'does the tile touch the block' |
| tile_overlaps_block | 'does the tile overlap the block' |
| tile_to_block | '' |
| extend_towards_until | 'extend tile towards a direction until the condition is met' |
| extend_towards_until_edge | 'extend tile towards a direction until it touches the edge' |
| extend_until_touches_block | 'extend tile towards a direction until it touches the edge' |
| move_towards_until | 'move tile towards direction until condition is met' |
| move_towards_until_edge | 'move tile towards direction until it touches edge' |
| move_until_touches_block | 'move tile towards direction until it touches block' |
| move_until_overlaps_block | 'move tile towards direction until it overlaps block' |
| get_tile_color | 'get the color of the tile' |
| tiles_to_blocks | '' |
| filter_tiles | 'only keep tiles that' |
| map_tiles | 'for every tile' |
| overlap_split_blocks | 'overlap the split blocks based on colors' |
| splitblocks_to_blocks | '' |
| color_logical | 'take logical operation on colors using them as true and false' |
| land | 'logical operator and' |
| lor | 'logical operator or' |
| lxor | 'logical operator xor' |
| negate_boolean | 'not' |
| map_tbs | 'for every block in template block scene' |
| make_colorpair | 'make pair of colors' |
| north | 'top' |
| south | 'bottom' |
| west | 'left' |
| east | 'right' |
| north_east | 'top right' |
| north_west | 'top left' |
| south_east | 'bottom right' |
| south_west | 'bottom left' |
| 0 | '0' |
| 1 | '1' |
| 2 | '2' |
| 3 | '3' |
| 4 | '4' |
| 5 | '5' |
| 6 | '6' |
| 7 | '7' |
| 8 | '8' |
| 9 | '9' |

| true | " |
|---|---|
| false | " |
| invisible | 'invisible' |
| black | 'black' |
| blue | 'blue' |
| red | 'red' |
| green | 'green' |
| yellow | 'yellow' |
| grey | 'grey' |
| pink | 'pink' |
| orange | 'orange' |
| teal | 'teal' |
| maroon | 'maroon' |

**Example Sampled Programs and Pseudoannotations**

| *Sampled Program* | *Natural Language Pseudoannotation* |
|---|---|
| (lambda (to_original_grid_overlay (remove_color (grid_to_block $0) yellow) false)) | 'place block on input grid remove color from block yellow' |
| (lambda (extend_towards_until_edge (block_to_tile (grid_to_block $0)) south_east) true)) | 'extend tile towards a direction until it touches the edge bottom right' |
| (lambda (blocks_to_min_grid (tiles_to_blocks (find_tiles_by_black_b $0)) true true)) | 'get the smallest grid containing the blocks find the tiles based on if they are separated by the black background' |

Compared to SCONE (Long et al., 2016), LARC poses a significantly greater challenge for distant supervision.

| | domain | dsl size | language kind | number of instances |
|---|---|---|---|---|
| LARC | DSL-open | 103 | freeform text | 354 |
| SCONE: ALCHEMY | DSL-closed | 24 | step-by-step instruction | 4560 |
| SCONE: TANGRAMS | DSL-closed | 14 | step-by-step instruction | 4989 |
| SCONE: SCENE | DSL-closed | 33 | step-by-step instruction | 4402 |

Table 3: Comparison of LARC to SCONE

### A.5 SUPPLEMENT TO SEC. 5: EXECUTING NATURAL PROGRAMS

**Enumeration details** For a task, we enumerate from the bi-gram distribution (proposed by the neural model) on a high-powered computing cluster for 720s; and with 24 CPUs in parallel.

**Other Models: Neural Sequence Decoder** We experiment with using a neural sequential decoder which can theoretically capture longer range dependencies. Specifically, we use GRU to decode a program one token at a time. In addition we mask the generated tokens to ensure the generated partial programs are syntactically correct (using the type system). We train using the distant supervision approach exactly as (Guu et al., 2017), with an epsilon-randomized beam search to balance exploiting the current policy and exploring low probability programs under the policy and take gradient steps on discovered programs using the meritocratic parameter update rule. We train using distant supervision on 24 CPUs for 10 hours of wall-clock time on the train split of 200 tasks. As we

| | Neural Sequence Decoder | |
|---|---|---|
| | training tasks discovered | testing tasks solved |
| IO | 6 / 200 | 2 / 183 |
| IO + NL | 7 / 200 | 0 / 183 |
| NL | - | 0 / 183 |

can see, the sequence decoder cannot even recover the 10 seed programs during training, and performs poorly on the testing tasks compared to the bigram model. Consequently, we did not attempt pseudo-annotation on the sequence model.

**Other Models: CNN encoding of IO** We take our best model (IO+NL+pseudo) and additionally condition the neural model with a CNN encoder, rather than leaving it un-conditioned. We find that this model can discover 2 more programs during training and achieves identical outcome to the simpler model without CNN.

| | train | test |
|---|---|---|
| IO+NL+pseudo | 21/200 | 22/183 |
| IO+NL+pseudo+CNN | 23/200 | 22/183 |

In general, we find that the standard solution to distant supervision, although effective in SCONE, only discovers a few programs in LARC. This finding is unsurprising for the following reasons:

1. LARC is DSL-open whereas SCONE is not, thus, there is *no* guarantee that we will discover all LARC programs even if we enumerate an *infinite* number of programs.

2. In SCONE, every computer program is a sequence of 5 actions that transform the state of the world. A natural language utterance is collected for each of these actions. The language annotation (natural program) is the sequences of these 5 utterances. As a result there a tight alignment from utterance to actions (tokens in the DSL).

3. SCONE domains have an order of magnitude more tasks to learn from (through distant supervision).

We conclude that collecting simpler, more fine-grained tasks as in SCONE would confer significant benefits to solving LARC, notwithstanding the DSL-open challenge.

## A.6 Synthesis with codex

We conduct a exploratory study where we took the tasks solved by the IO+NL specification in addition to the IO-only specification, and see whether github's co-pilot auto-complete tool (built on codex) can correctly infer the right program using language as prompt. The prompt is constructed by giving a few hundred of pseudo-annotation - program pairs as context (see A.4), along with the NL description from LARC, and asking co-pilot to give the top-10 suggestions on how this description might be translated into a program in our DSL.

| tasks | NL | correct program | example copilot program | # of syntactically valid out of 10 samples | # of correct |
|-------|-----|-----------------|-------------------------|---------|---------|
| 0520fde7.json | look at both the left and right parts of the input grid. 'You will notice that the left and right parts are 3x3. For each square that is colored on both the left and right parts, color the output grid with red on the new 3x3. | (lambda (overlap_split_blocks (split_grid $0 false) (lambda (lambda (color_logical $0 $1 red land))))) | (lambda (to_original_grid_overlay (grid_to_block $0) (is_symmetrical (grid_to_block $0) true))) | 2 | 0 |
| 0b148d64.json | copy the pattern that is a different color | (lambda (to_min_grid (remove_color (remove_black_b (grid_to_block $0)) (nth_primary_color (grid_to_block $0) 1)) false)) | (lambda (to_original_grid_overlay (grid_to_block $0) (has_color (grid_to_block $0) pink))) | 4 | 0 |
| 1f85a75f.json | Copy only the biggest shape into the output grid | (lambda (to_min_grid (first_of_sorted_object_list (find_same_color_blocks $0 true false) (lambda (get_num_tiles $0)) false) false)) | lambda (copy_shape_to_output_grid (first_of_sorted_object_list (find_same_color_blocks $0 false false) (lambda $0) true)) | 0 | - |
| 23b5c85d.json | change the grid size with the colored object with the smallest area and fill the same color | (lambda (to_min_grid (first_of_sorted_object_list (find_same_color_blocks $0 true false) (lambda (get_num_tiles $0)) (negate_boolean false)) false)) | (lambda (to_original_grid_overlay (fill_color (grid_to_block $0) (grid_to_block $0)) true))) | 1 | 0 |
| 5582e5ca.json | determine which color is used for the majority of the squares in the input grid, then use that color for the entire output grid. | (lambda (to_original_grid_overlay (fill_color (merge_blocks (find_blocks_by_inferred_b $0 true false) true) (nth_primary_color (grid_to_block $0) 0)) true)) | (lambda (to_original_grid_overlay (find_majority_color $0) (lambda (lambda $0)))) | 0 | - |
| 6f8cd79b.json | create a light blue border around the grid. So, any squares that are touching the edges are filled in with light blue. The rest of the squares inside the light blue border remain black | (lambda (to_original_grid_overlay (fill_color (filter_block_tiles (grid_to_block $0) (lambda (is_exterior $0 false))) teal) true)) | (lambda (to_original_grid_overlay (map_block_tiles (grid_to_block $0) (lambda (block_to_tile (set_color (reflect (grid_to_block $1) true) light_blue)))) true)) | 0 | - |
| 72ca375d.json | copy one of the figures and just that figure. | (lambda (to_min_grid (singleton_block (filter_blocks (find_same_color_blocks $0 true false) (lambda (is_symmetrical $0 false)))) false)) | (lambda (to_original_grid_overlay (copy_grid $0) (lambda (lambda (has_color (fill_color (grid_to_block $0) black)))))) | 0 | - |

Figure 12: synthesizing programs using copilot

As we can see, while co-pilot suggests programs that look similar to a correct one stylistically, most tend to be syntactically invalid. Copilot often invents primitives that do not even exist in our DSL, such as "copy_shape_to_output_grid". None of the syntactically correct programs can produce the intended output either. This is to be expected, as we use a DSL that has not been seen before in any existing corpus of code, and we should not expect codex to perform well naively. We believe a promising line of work lies in taking a general model (such as codex) and specializing it to a specific context (LARC) will be exciting future research.

## B APPENDIX : MULTI-BANDIT, INFINITE-ARM, BEST-ARM IDENTIFICATION

Imagine there are N different mAgIcAl casinos, where each has an infinite number of slot machines (arms). While each individual arm has its own probability $p$ (Bernoulli) of generating an outcome of either 0 or 1, the arms are related to each other depending on the casinos they belong to. Some casinos are easier than others, in a sense that for some, it is easier to find a "good" arm whereas for others, most arms will have a small chance of success. Moreover, each casino $i$ has one (or multiple) best arm, whose probability of generating a 1 is $p_i^*$. Your job is to identify the best arm within each casino. This is in essence the multi-bandit, infinite-arm, best-arm identification problem.

You can take observations in the casinos, where each observation involves selecting a casino, and trying one of its arms (either one of the arms you already tried, or trying a new one out of its infinite possibilities), observing an outcome of either 0 or 1. We seek an online algorithm that, given any observation budget, propose a set of N arms. Let $p_1 \ldots p_N$ denote the ground-truth Bernoulli parameters of the proposed arms. We seek to minimize the following regret:

$$L = \sum_i (p_i^* - p_i)$$

Where each term $p_i^* - p_i$ is the "gap" between the proposed arm and the best arm in a given casino.

### B.1 APPLICATION TO LARC

Our goal is to collect a working natural program for each of the 400 ARC tasks. Natural programs are difficult to collect, because it involves both: 1) obtaining a natural program from a describer and 2) validating this natural program by having a builder build from it. Thus, rather than exhaustively studying each task to estimate its difficulty, we are content with just getting a "good enough" natural program for each task. In another words, given a certain annotation budget, we want to find a single good natural program for each of the 400 tasks.

If we take the 400 tasks as 400 casinos, then each casino would have an intrinsic difficulty, which corresponds to how easy it is to communicate a particular task. Within each task, there are an infinitely many possible natural programs (i.e. all natural language strings), which correspond to the infinite-arm aspect. For each task, we are interested in finding as good of a description as we can, which correspond to the best-arm identification aspect.

Specifically, we are seeking an online algorithm that *at any budget* can propose a set of natural programs, and this set of proposed programs should improve with added budget (budget here is synonymous with total participants' time). To use the bandit algorithm in conjunction with the annotation process, we divide the 45 minutes of a participant's time into several "units" of participation, where each unit can be assigned to one of two jobs: 1) The participant can either give a new description to an ARC task, then immediately build from it (in the form of describer verification) or 2) The participant can be given an existing description of a task, and build from it to to assess if it is a good description. See Figure 13. We estimate how many minutes would this particular unit take, and dynamically allocate additional units until the full 45 minutes are exhausted.

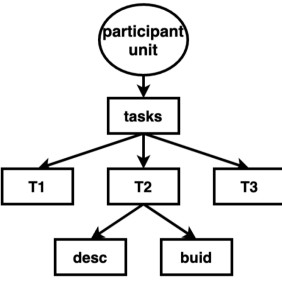

Figure 13: How a "unit" of a participant's time can be utilized

## B.2 REINFORCEMENT LEARNING FORMULATION

A great way to formalize a bandit problem is casting it as an instance of a Markov Decision Process:

A **state** consists of all the observations (the 0, 1 outcomes) on all the arms thus far. Let there be $N$ bandits/casinos, then the observation is a collection of all casinos' outcomes $C_1 \ldots C_N$ where for each casino $C_i$, we have observation for its $K$ arms that we already sampled: $c_i^1 \ldots c_i^K$. Each arm's observation, $c_i^j$ is simply a tuple $(A, B)$ where $A$ denotes the number of 1s observed from arm $c_i^j$ and $B$ denotes the number of 0s. Thus, the space of observation is $O(N \times K \times (A + B))$. See Figure 14.

There are two kinds of **actions** – the *arm-selection* action, and the *best-arm-proposal* action. Arm selection consists of a tuple $(i, j)$ where $i$ selects a casino, and $j$ selects from which of the arms within that casino to sample an additional observation. We will use $j = 1 \ldots K$ to denote sampling from the $K$ arms within a particular bandit $i$, and use $j = 0$ to denote sampling a *new* arm from bandit $i$. When the interaction budget is exhausted, the agent must make a best-arm-proposal action, in which the agent picks one sampled arm from each casino to be calculated in the regret. For arm proposal, we use a simple heuristic that selects the arm with the highest estimated mean using a beta distribution with (1,1) prior. For the remainder of this section, **action** will refer exclusively to arm-selection.

**Transition**   modifies the **state** to include the new observation. See Figure 14.

**Reward**   is the sum of the Bernoulli parameters for the set of proposed arms. $p_1 + \cdots + p_N$.

```
original state:

[(2,0), (1,1), (0,1)]   # casino1, has 3 arms, 3rd arm has 0 success and 1 fail

[(1,1), (1,0)]          # casino2, has 2 arms, first arm has 1 success and 1 fail

[ ]                     # casino3, has 0 arms so-far

taking the 2 following actions:

action (3,0) observed a 0, and action (2,1) observed a 1

resulting state after those 2 actions:

[(2,0), (1,1), (0,1)]   # casino1, has 3 arms, 3rd arm has 0 success and 1 fail

[(2,1), (1,0)]          # casino2, has 2 arms, first arm has 2 success and 1 fail

[(0,1)]                 # casino3, has 1 arm, first arm has 0 success and 1 fail
```

Figure 14: an example transition where there are 3 casinos

## B.3 A HEURISTICALLY DEFINED AGENT

To the best of our knowledge, there is no bandit algorithm that address the specific bandit problem we are solving. However Wang et al. (2008) solves the infinitely many armed bandit problem for a single bandit, where they explicitly model the difficulty of the underlying bandit. We take their algorithm as inspiration. Note that Wang et al. (2008) prescribe a solution to the regret-minimization problem, which is not exactly best-arm-identification. However, in the limit, the two are equivalent as minimizing regret is equivalent to finding the optimal arm. We will first state the result of Wang et al. (2008), which applies to the case of a single casino/bandit, then extend it to the case of multi-bandit.

**arm selection**   Suppose we know that we want to generate an action in casino $i$. Wang et al. (2008) proposed the following rule for selecting which arm to interact with. Let $\beta$ be the difficulty parameter of the task, defined as: $P(p^* - p_j < \epsilon) = \Theta(\epsilon^\beta)$. Which is to say, if you were to sample

a new arm with ground truth parameter $p_j$, the probability that this arm lies within $\epsilon$ of the optimal arm, is approximately $\epsilon^\beta$. For instance, if $\beta = 1$, the task is very difficult as $\epsilon^1$ is a tiny number, meaning it is almost impossible for you to sample an arm $p_j$ that is $\epsilon$ close to optimum. Conversely, if $\beta = 0$, the task is very simple, as $\epsilon^0 = 1$, so any arm you sample will be optimal.

Wang et al. (2008) states that, if you let $M$ be the total number of observations on a bandit, and $K$ be the total number of arms currently sampled, if $K \leq M^\beta$, then you should sample a new arm. Otherwise, you should perform the standard UCB algorithm on the set of existing arms. In our bandit RL environment, $M$ and $K$ are well defined, but how do we estimate $\beta$? We use the following heuristic to estimate difficulty: Let $j$ be the best arm in the current casino w.r.t. its sampled mean $\tilde{p}_j$, then we define $\beta = 1 - \tilde{p}_j$. For instance, if the best arm has a sampled mean of 0.9, then we are in an "easy" casino, and the difficulty will be $1 - 0.9 = 0.1$, which is fairly close to 0, implying we should *NOT* be sampling new arms, as the best arm we have currently is likely to be good. Conversely, if the best arm has a sampled mean of 0.1, then we are in a "difficult" casino, where we stand a better chance of finding a good arm by sampling more arms.

**casino selection** To adopt the infinitely-many arm algorithm to a multi-bandit setting, we use the following heuristic: selecting the casino where we have the least information about $p^*$ of a casino. In practice, we rank all $K$ arms based on their sampled mean, and take the top-half of the arms, and aggregate a beta distribution of the total number of 1s and 0s of these arms, and use the variance of the beta distribution as a proxy for uncertainty. For instance, if a casino whose top-half arms have in total many observations, and most of them are 1s, then we are certain about its $p^*$. Conversely, if a casino whose top-half arms have few observations, and it is an even split of 1s and 0s, we are unsure of its $p^*$.

### B.4 SIMULATED EVALUATION

With both arm selection and casino selection, we have a functioning agent. We can evaluate this agents' performance against several baseline agents in the bandit RL environment to verify that it is indeed more efficient. We consider the following baseline agents, **rand** is the random agent that select an action at random, **tile** is the agent that tries to evenly spread out the observation budget, **tile-inf** is the agent that uses the infinitely many arm algorithm, and tries to spread the budget evenly across casinos, **cas-inf(ours)** is the agent that selects the casino using uncertainty of $p^*$, and use infinitely many arm algorithm.

The algorithms performance over 100 casinos with a total of 600 interaction budgets is in Figure 15

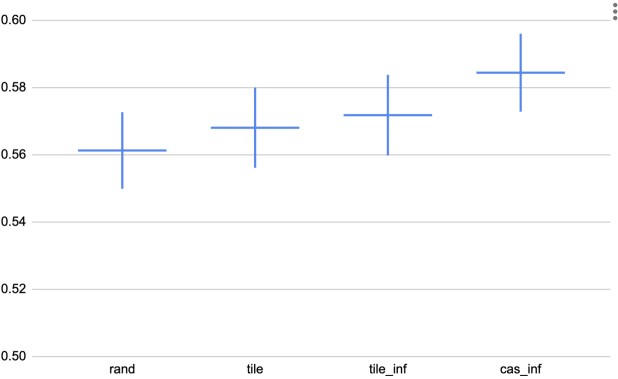

Figure 15: performance of various bandit policies, of 100 casinos and a budget of 600, averaged across 100 repetitions. horizontal bar is average, whiskers indicate standard deviation

As one can see, for the simulated environment, which makes several simplifications, such as not taking in the generation aspect of description making, and modeling difficulty of a casino as a truncated gaussian, our proposed bandit algorithm out-performs the other baselines. The implementation of the bandit environment and the bandit policies can be found at `LARC/bandit`

