# OpenReview forum: "Communicating Natural Programs to Humans and Machines"
_ICLR.cc/2022/Conference — ICLR 2022 Submitted_

### Official Review · Reviewer_iQUC · 2021-10-26

**Correctness:** 4
**Technical Novelty And Significance:** 1
**Empirical Novelty And Significance:** 2
**Recommendation:** 5
**Confidence:** 4

**Main Review:**

### Strengths:
- The authors are attempting to tackle an important program in artificial intelligence: how to tackle open-ended puzzle learning, communication, and resolution. Past approaches have selected fixed goal posts via DSLs, implying there exists a known solution to the problem, while here the authors build off of ARC which has no DSL, and thus could prove to be unreachable via any standard DSL. Fortunately the authors show that in 88% of the cases human descriptions of a solution can correctly communicate a solution to another human. This single observation is one of the key achievements of the paper, and is interesting enough in that it constructs a new artifact that machines could try to approximate instead of tackling ARC directly.
- The author methodology for obtaining the samples is cost efficient, and provides good coverage over the dataset.
- The proposed approach is original and could prove valuable to researches in AI communication, planning, knowledge representation, linguistics, and meta or multimodal learning.

### Weaknesses:
- While the contribution of a natural language caption over the ARC dataset is compelling, is it not yet clear whether this data can prove useful for others downstream. The inclusion of a seq2seq model accuracy at predicting these descriptions, or some other form of evaluation would help highlight whether these annotations are within reach of current approaches, or too diverse or unpredictable to be effectively captured by present conditional language models.
- Existence proof of the usability of a description certainly proves that some humans can save a task, but does not demonstrate the robustness of a description to all human receivers — in this sense it might be useful to either describe this limitation more extensively, provide an example, or a counter-example of this shortcoming.


**Summary Of The Paper:**

Authors construct a dataset of natural language programs for solving ARC tasks by using a population of Mechanical Turk workers to validate that a turker can correctly communicate how to solve an ARC task to another turk purely via the natural language used. The collected dataset is then analysed by the author and evidence is gathered for the richness of the programming primitives implied by natural language, the use of refining language, and the diversity of terms. The authors use this analysis to motivate the need for an improved DSL for solving ARC.

**Summary Of The Review:**

Paper presents a useful dataset complementary to the ARC effort for understanding the limits of DSLs and capabilities of natural language for expressing programs. The dataset appears to be adequately constructed, however a lack of downstream applications and proof of utility make the effort harder to contextualise, and the dataset can be seen as incremental over the original ARC effort.

---

> ### Author Response · Authors · 2021-11-16
> **response to R3**
>
> “The inclusion of a seq2seq model accuracy at predicting these descriptions”
>
> - We are uncertain what seq2seq mean in this set up, i.e. is the input and what is the output for such seq2seq model? If the input sequence is NL descriptions, and the output sequence is code, it is exactly the setup for our synthesis section. If we want to generate these descriptions from the input-output images, a possible venue might be image captioning, which we do not believe will perform well with only ~500 paired annotations that are highly precise in nature -- requiring one to solve the puzzle encoded in image form. That is to say, the current language descriptions will be too diverse and unpredictable for current language models.
>
> “Existence proof of the usability of a description certainly proves that some humans can save a task, but does not demonstrate the robustness of a description to all human receivers — in this sense it might be useful to either describe this limitation more extensively, provide an example, or a counter-example of this shortcoming.”
>
> - Our bandit algorithm (appendix B) is explicitly constructed with the property that:
>
> 1. each task gets a fair share at having a working description being proposed
> 2. of the current working descriptions, validate that it is robust under multiple receivers
> 3. do 1 and 2 under cost constraints
>
> In general, it will be ideal if we can extensively validate all descriptions using many builders, but this will come at significant cost. The generative nature of the ARC tasks also guards against the builder getting the right answer “by chance”, as the correct outcome requires one to generate a pixel-perfect output grid.
> However, we agree it is important to study how builders may fail to build from a description. For instance, a trend we observed is that the describer often leaves out key information such as re-coloring a block as the last step, causing the builder to fail. Or studying how a working description for 1 builder fails on another builder.

---

### Official Review · Reviewer_4uxi · 2021-11-02

**Correctness:** 3
**Technical Novelty And Significance:** 3
**Empirical Novelty And Significance:** 3
**Recommendation:** 5
**Confidence:** 4

**Main Review:**

*Strengths*
- The paper addresses an important problem: creating a benchmark aimed towards creating machine learning models that can help understand and reason concepts. Adding language annotations ( which is the way humans communicate) to ARC is the right direction to progress especially keeping in view the surge of large langauge models for both natural language and code.
- The paper is well written in most parts ( especially introduction and motivation). However, there are significant improvements that can be done in terms of improving the clarity ( see comments below).

*Weaknesses*
- Even though the motivations are well set, I think just adding langauge to an existing dataset might be considered less novel. The linguistic analysis is interesting but I would have liked to see the results of more program synthesis systems especially the large models like Codex [1], GPT-Neo [2], GPT-J [3]. I feel the addition of language naturally makes the set of tasks in LARC more suitable for such language models that makes it essential to have this comparison. Using a program synthesis system that is not catered towards accepted natural langauge instructions (the system used in the paper) will not lead to significant improvements.
- I felt that the clarity of the paper needs to be improved ( see concrete suggestions below). On a high-level including more examples will really help in understanding the ideas put forth in the paper.


*Suggestions/ Comments*

1. An example in each category of the tag will be useful. Just describing the results with these tags used directly in the sentences seemed a bit abrupt to me.
2. More details should be provided about the exact program synthesis model employed.
3. I felt the bandit algorithm for data collection was interesting and if more details about it are placed in the main paper, it will definitely add much value.
4. More description of how psedo-annotations are produced along with examples.
5.  More description of how exactly the distant supervision algorithm works.
6. I found the brief section on suggestions quite interesting. Personally, I would like if the authors could expand this section with more observations drawn from other program synthesis systems and evaluation settings.
7. Some sample cases showing in which cases the program synthesis system succeeds and why will be extremely useful. In fact, I will say it is needed to understand the role and contributions brought in by adding language. It might be possible that the DSL itself is not defined properly or the language annotations or pseudo-annotations are not good enough to accomplish the set of reasoning required to solve the particular task. Basically, more analysis of why the systems fail so terribly through sample cases needs to be added.

[1] https://openai.com/blog/openai-codex/
[2] https://github.com/hendrycks/apps
[3] https://arxiv.org/abs/2009.03393

**Summary Of The Paper:**

The paper extends the ARC dataset by adding human-written instructions that they term as "natural programs".  The extended dataset is called LARC. They draw parallels and cite differences between computer programs and natural programs. They draw some linguistic insights based on the natural programs and finally study the performance of a program synthesis model on the LARC dataset.

**Summary Of The Review:**

The paper studies an important problem and the motivation is well laid. However, I feel the paper needs some work in terms of improving clarity as well as more extensive evaluation of large language models. They also need failure case analysis elucidating some sample cases.

---

> ### Author Response · Authors · 2021-11-16
> **response to R2**
>
> “I think just adding langauge to an existing dataset might be considered less novel.”
> - This assessment would be correct, if we are merely augmenting ARC with language. However, we are adding language in a specific way:
> “language-complete instructions: they can be demonstrably interpreted by other human users to correctly produce the intended outputs without any additional contexts (including the original input-output examples).”
> Thus, one should think of the language in LARC not as accompanying the original IO specifications, but as a complete reformulation of the original IO specification that is self-sufficient. This is crucial for the validity of our analysis of LARC programs:
> “Since LARC is language-complete, analyzing the words used in LARC serves as a good proxy for the underlying concepts present in the ARC domain.”
> This way, we can analyze the concepts within ARC entirely from a linguistic point of view, without fearing that certain concepts are left-out of the description.
>
>
> “I would have liked to see the results of more program synthesis systems especially the large models like Codex [1], GPT-Neo [2], GPT-J [3].”
> - We do use a large language model, T5, in our language-guided synthesis. We have also conducted an explroatory study, see updated revision, see section A.6
>
>
> “Using a program synthesis system that is not catered towards accepted natural language instructions (the system used in the paper)”
> - We respectfully disagree. The distant supervision approach [1] is the approach directly catered towards leveraging language information to guide program synthesis for our problem.
>
> “An example in each category of the tag will be useful. Just describing the results with these tags used directly in the sentences seemed a bit abrupt to me.”
> - Space is limited in the main paper, thus we left it to A.3, with multiple examples per tag.
>
> “More details should be provided about the exact program synthesis model employed.”
> - More details are given in A.4 and A.5, we will also include an algorithm block to make distant supervision (outlined in [1]) more clear.
>
> “I felt the bandit algorithm for data collection was interesting and if more details about it are placed in the main paper, it will definitely add much value.”
> - We are quite fond of the bandit algorithm as well! However, given the space constraint of the main paper, we believe that explaining the implications of natural programs to be more important. We are glad you enjoyed the bandit section.
>
> “More description of how psedo-annotations are produced along with examples.”
> - See A.4
>
> “More description of how exactly the distant supervision algorithm works.”
> - Will add algorithm block for this.
>
> “I found the brief section on suggestions quite interesting. Personally, I would like if the authors could expand this section with more observations drawn from other program synthesis systems and evaluation settings.”
> - Thanks for suggesting this, the additional exploratory exercise with co-pilot was quite informative.
>
> “Some sample cases showing in which cases the program synthesis system succeeds and why will be extremely useful. In fact, I will say it is needed to understand the role and contributions brought in by adding language. It might be possible that the DSL itself is not defined properly or the language annotations or pseudo-annotations are not good enough to accomplish the set of reasoning required to solve the particular task. Basically, more analysis of why the systems fail so terribly through sample cases needs to be added.”
> - Program synthesis system succeeds when it can correctly infer a program given the search budget, and its performance improves if the correct program has higher likelihood of being searched given the generative model. We study this in Figure 9, where the addition of language improves the likelihood of sampling the correct primitives. This improvement hinges on the assumption that program-language follows a close 1-1 mapping, which is not true in LARC, as indicated by the linguistic analysis section (sec 4).
>
> [1] From language to programs: Bridging reinforcement learning and maximum marginal likelihood.

---

### Official Review · Reviewer_doKU · 2021-11-03

**Correctness:** 3
**Technical Novelty And Significance:** 2
**Empirical Novelty And Significance:** 3
**Recommendation:** 5
**Confidence:** 3

**Main Review:**

Strengths:

1. The paper is well-motivated with the goal of studying the gap between communicating natural programs to humans and machines. The existing reasoning benchmark shows there still exists a large gap in reasoning and problem solving between them. It's a critical problem to be addressed.

2. The LARC dataset construction is by collecting verifiable instructions from a describer and builder, which ensures the task finish rate. The dataset is valuable for the community to study how natural program boosts the abstraction and reasoning process. The linguistic analysis shows how concepts differ in computer and natural programs.

3. It uses existing program synthesis techniques to study how to execute the natural programs as humans. Showing solid qualitative and quantitative findings.

Weaknesses:

1. The technical contribution of this paper is weak since it mainly uses previous approaches to analyze the program synthesis results. I would like to see a new learning method to tackle this challenge.

2. The improvement from 16/183 to 22/183 is not trivial but still very poor. One main reason is the DSL-based program synthesis is limited by the scale and range of the DSL. The analysis of how the scale and coverage range of DSL affects the performance is required for a better understanding of how to close the gap.

**Summary Of The Paper:**

The paper presents the language-complete ARC to study how human language can affect abstraction and reasoning capability.  It collects the natural program described by human annotators and applies program synthesis techniques to analyze the gap between communicating to humans and machines. The motivation is sound and the contribution is solid with the effort of annotating LARC and benchmarking the program synthesis approaches.

**Summary Of The Review:**

I like the idea and solid contribution of this paper. The main concern I have is the technical contributions. Please refer to the above for details. I would consider raising the score if the concern is properly addressed.

While I am not totally familiar with all the literature, my assessment of the novelty and originality might not be accurate enough.

---

> ### Author Response · Authors · 2021-11-16
> **response to R1**
>
>
> “The technical contribution of this paper is weak since it mainly uses previous approaches to analyze the program synthesis results. I would like to see a new learning method to tackle this challenge.”
>
> - As stated in the overall response, such a method will require significant work beyond the current synthesis capabilities, which we believe is out of scope for this work.
>
>
> “The improvement from 16/183 to 22/183 is not trivial but still very poor.”
>
> - Indeed, as we are interested in applying standard baseline algorithms, and we do not expect drastic improvements with the addition of language (although we do expect, and observe, some improvements).
>
>
> “One main reason is the DSL-based program synthesis is limited by the scale and range of the DSL. The analysis of how the scale and coverage range of DSL affects the performance is required for a better understanding of how to close the gap.”
>
> - A full description of our DSL can be found in (A.4), highlighting the extent the attempted LARC DSL is similar/different from SCONE, the closest synthesis benchmark. In general, the coverage problem of a DSL (i.e. given a task, decide whether there is a program within the DSL that solves the task) is intractable, because it requires one to construct all possible programs within a DSL, and checking if any of them is correct for a given task. A more appropriate measure would be “effective coverage under synthesis”, which treats tasks that cannot be synthesized to be “effectively out of the DSL”. Under this definition, we could say 161/183 tasks are outside the scope of our current DSL, as there are no ground-truth programs for these ARC tasks to begin with. Clearly, running the synthesizer with more computational power will shrink the number of tasks outside the scope, and having a turing-complete DSL would imply that no tasks will be outside the scope, but is not practical.
>
> One could imagine a study where we take the current attempted DSL, and remove at random some primitives from it, and re-run the synthesis experiment. We should expect a decrease in the number of programs solved, as they would be “out-of-scope”, but this setup is not very insightful.
>
> Alternatively, one could build several different DSLs (induced over different subsets of ARC) and see if they generalize across these divisions. We agree this is a good idea, and will take a serious attempt at it from a more community-driven fashion (similar to how LARC dataset was curated), as we believe the current approach of having a single designer writing the entirety of a DSL to be prohibitively expensive.
>
> Overall, we agree with this suggestion, but feel it will be very expensive to evaluate it thoroughly at this point. If you have any good ideas we would love to hear them, as we agree strongly that measuring coverage is important for this line of research.

---

### Author Response · Authors · 2021-11-16
**re : this work does not contain a novel algorithm that significantly outperforms the baseline algorithms**

We will use this top level response to address concerns that are common across reviewers, reserving the individual response for more specific queries.

The main weakness of the paper, as pointed out by the reviewers, is that it does not contain a novel algorithm that significantly outperforms the baseline algorithms by leveraging the language annotations.

“I would like to see a new learning method to tackle this challenge.” -- R1

“I would have liked to see the results of more program synthesis systems especially the large models like Codex” -- R2

“While the contribution of a natural language caption over the ARC dataset is compelling, is it not yet clear whether this data can prove useful for others downstream.” -- R3

We believe that highlighting the difference of natural vs computer programs in a DSL-open synthesis domain, through extensive annotation efforts and linguistic analysis, along with  recommendation for future synthesis systems grounded in these analyses to be adequate for a typical conference submission.

While developing a new learning method to leverage these findings is definitely important, and the ultimate goal for this line of research, given the intrinsic difficulty of ARC, it will be a multi-year collaborative effort across different communities. We trust that our work is the first step in the right direction, by making it explicit _why_ ARC is challenging for existing approaches, and outlining specific sub-goals -- that of growing and referencing an unbounded number of concepts and communicating beyond paraphrasing procedures -- that must be addressed in order to solve ARC.

Finally, we would like to make clear that ARC is one of the most challenging program synthesis benchmarks up to date, as it is not constructed to demonstrate the efficacy of a newly proposed algorithm, but constructed to be a challenge, out of reach of current approaches. Consequently, there are not many efforts in tackling ARC, as small modifications to existing approaches will not yield significant improvements. We believe that while immediate improvements in solving ARC might be slow, it does not diminish the fact that ARC is an important challenge that one should make an honest attempt in solving in the long run.

---

### Decision · Program_Chairs · 2022-01-20

**Decision:**

Reject

**Comment:**

The paper presents the Language-complete Abstraction and Reasoning Corpus (LARC): a collection of natural language descriptions by a group of human participants who instruct each other on how to solve tasks in the Abstraction and Reasoning Corpus (ARC).

Overall, the reviewers found the LARC benchmarks to be well-motivated. However, there were concerns about whether the value of the dataset to downstream tasks. Results from additional program synthesis systems (like Codex and GPT-Neo) would also make the paper stronger. I agree with these objections and am recommending rejection this time around. However, I encourage the authors to continue pursuing this line of work and resubmit after incorporating the feedback from this round.